# Crystal Chemistry of Barian Titanian Phlogopite from a Lamprophyre of the Gargano Promontory (Apulia, Southern Italy)

**Gennaro Ventruti** [1,*] **, Alfredo Caggianelli** [1] **, Vincenzo Festa** [1] **and Antonio Langone** [2]

1. Dipartimento di Scienze della Terra e Geoambientali, University of Bari, Via Orabona 4, I 70125 Bari, Italy; alfredo.caggianelli@uniba.it (A.C.); vincenzo.festa@uniba.it (V.F.)
2. CNR-Istituto di Geoscienze e Georisorse, sezione di Pavia, via Ferrata, 1, I 27100 Pavia, Italy; antonio.langone@igg.cnr.it
* Correspondence: gennaro.ventruti@uniba.it

**Abstract:** This study is focused on a barian titanian phlogopite found in an alkaline ultramafic dyke transecting Mesozoic limestones of the Gargano Promontory (Apulia, Italy). The rock containing the barian titanian phlogopite, an olivine-clinopyroxene-rich lamprophyre with nepheline and free of feldspars, has been classified as monchiquite. The present study combines chemical analyses, single crystal X-ray diffraction and Raman spectroscopy. Chemical variations suggest that the entry of Ba into the phlogopite structure can be explained by the exchange Ba + Al = K + Si. The crystal structure refinement indicates that the Ti uptake is consistent with the Ti–oxy exchange mechanism. The structural parameters associated with the oxy substitution mechanism are extremely enhanced and rarely reported in natural phlogopite: (a) displacement of M2 cation toward the O4 site (~0.7); (b) M2 octahedron bond-length distortion (~2.5); (c) very short $c$ cell parameter (~10.14 Å). Raman analysis showed most prominent features in the 800–200 cm$^{-1}$ region with the strongest peaks occurring at 773 and 735 cm$^{-1}$. Only a weak, broad band was observed to occur in the OH-stretching region. As concerns the origin of the barian titanian phlogopite, the rock textural features clearly indicate that it crystallized from pockets of the interstitial melt. Here, Ba and Ti enrichment took place after major crystallization of olivine under fast-cooling conditions, close to the dyke margin.

**Keywords:** Ti–Ba-bearing phlogopite; Ti–oxy substitution; single-crystal structure refinement; Gargano promontory; Raman spectroscopy; lamprophyre

## 1. Introduction

Phlogopite is a rock-forming mineral that plays a key function in geochemical and petrological processes of the upper mantle [1]. In this respect, the role of some elements such as Ti and Ba is important. It was found that Ti content critically expands the thermal stability of phlogopite, consequently affecting melting processes within the upper mantle [2]. The experimental study of Guo and Green [3] showed that a Ti increase is accompanied also by an increase in Ba content, concluding that the genesis of ultrapotassic lamproitic magmas can be related to partial melting of phlogopite-bearing peridotite.

The increase of the Ti content in phlogopite has a relevant impact also on technological applications. The entry of Ti into the structure improves the electronic conductivity, an important factor in the fabrication of heterostructures, combining with mica nanosheets to yield a variety of novel and interesting optical and electrical properties [4].

The discovery of a phlogopite with an unusual high Ti and Ba content in an alkaline ultramafic dyke of the Gargano Promontory in the Apulia region of southern Italy (Figure 1a) prompted us to

undertake a crystal chemistry and petrological study. Although quite rare, barian titanian phlogopites and biotites have been identified in a wide range of magmatic lithotypes such as gabbro, [5–7], ultramafic rocks [8,9], nephelinite [10,11], leucitic rocks [12,13], melilitite [14,15] and lamprophyre [16].

Structural features, for instance bond-lengths and distortion parameters, are remarkably different when Ti and Ba are incorporated in the phlogopite structure [17,18].

The interlayer cavity adjusts in size in order to allow Ba to produce more hexagonal shapes of the tetrahedral rings as a consequence of the preference of the divalent Ba to surround itself by as many neighbors as possible [19–21]. $Ba^{2+}$-for-$K^{1+}$ substitution in the interlayer site requires a charge compensation that can involve different cations in tetrahedral coordination and/or vacancies in the interlayer site [22]. Ba–K substitution mechanisms can be easily tested by representing chemical variability plots of Ba vs. vacancies or Ba + Al vs. K + Si.

The main exchange mechanisms by which Ti enters the phlogopite octahedral layer have been reviewed in detail in Waters and Charnley [23] and generally involve the occurrence of vacancies, oxidation, fluorination or coupled substitutions in different sites. In most cases, Ti substitution schemes in barian titanian phlogopite have been determined only by means of correlation plots on the basis of microprobe analyses, and this may lead to a misinterpretation of the correct exchange mechanism occurring in the structure when structural details are lacking or when data on $Fe^{3+}$ and OH content are not available. Thus, when Ti–oxy substitution is not properly verified, the Ti uptake is generally modeled by the Ti-vacancy exchange [24,25].

Clear evidence for the Ti–oxy substitution has been recognized by means of single-crystal X-ray diffraction [17] in absence of $Fe^{3+}$ and/or $OH^-$ determinations. According to these Authors [17] the shortening of the *c* lattice parameter coupled to the marked distortion of M2 octahedron, where Ti is preferentially partitioned, are strong indications of the oxy mechanisms.

Crystal structure refinements on biotite, fully characterized in terms of anion and cation speciation, have validated the relationships between the Ti–oxy mechanism and key structural features [26]. Recently, other structural parameters have been further introduced to better assess the main Ti-substitution mechanism [27,28].

In this work, after an introductory part devoted to outlining the geological setting and to define the nature of the hosting magmatic rock, a characterization of a barian titanian phlogopite is presented. To this end, we performed an approach that includes a petrographic analysis, electron microprobe analysis (EMPA), scanning electron microscopy coupled with energy dispersive spectrometer (SEM-EDS), single-crystal X-ray diffraction (SCXRD) and micro-Raman spectroscopy. Inductively coupled plasma mass spectrometry (ICP-MS) for whole rock analysis was also carried out. In addition, environmental conditions favoring crystallization of the studied barian titanian phlogopite are also examined.

## 2. Geological Setting

The Gargano Promontory is the site of two distinct occurrences of mafic to ultramafic intrusive rocks of alkaline affinity. These can be found at Punta delle Pietre Nere, close to the Lesina Lake and at San Giovanni in Pane locality, in the Apricena territory (Figure 1a). The melasyenite and melagabbro magmatic body of Punta delle Pietre Nere—brought up to the surface by the Lesina gypsum diapir [29]—have been the object of several studies that defined an age of 62–56 Ma [30,31] and compositional features typical of lamprophyric rocks [32], in the Apulian foreland domain [33]. Vollmer [30,34] related the within-plate magmatism of Punta delle Pietre Nere to the activation of an asthenosphere hotspot, later responsible for the genesis of the Roman Comagmatic Province volcanism. In terms of isotopic composition, the Punta delle Pietre Nere magmas show an intraplate signature free of subduction influence [33,35], acquired from a mantle source located in the neighborhood of the spinel- to garnet-peridotite transition [36].

A decidedly lower number of studies is available for the magmatic rocks of San Giovanni in Pane, close to a former Monastery (41°47′37″ N; 15°23′40″ E) and approximately 15 km south of the Punta delle Pietre Nere outcrop (Figure 1a). In the nineteenth century, Pilla [37] first described these

enigmatic magmatic rocks, which were subsequently interpreted by Carella [38] as constituting an E–W oriented, 40-m-thick dyke intruding Cretaceous limestones (Figure 1b). The first analyses of the San Giovanni in Pane rocks were provided by Amendolagine et al. [39] that also performed a comparative study with the Punte delle Pietre Nere rocks. In recent surveys of the zone—subsequent to intensive quarry excavation—only limited blocks of the former dyke remain.

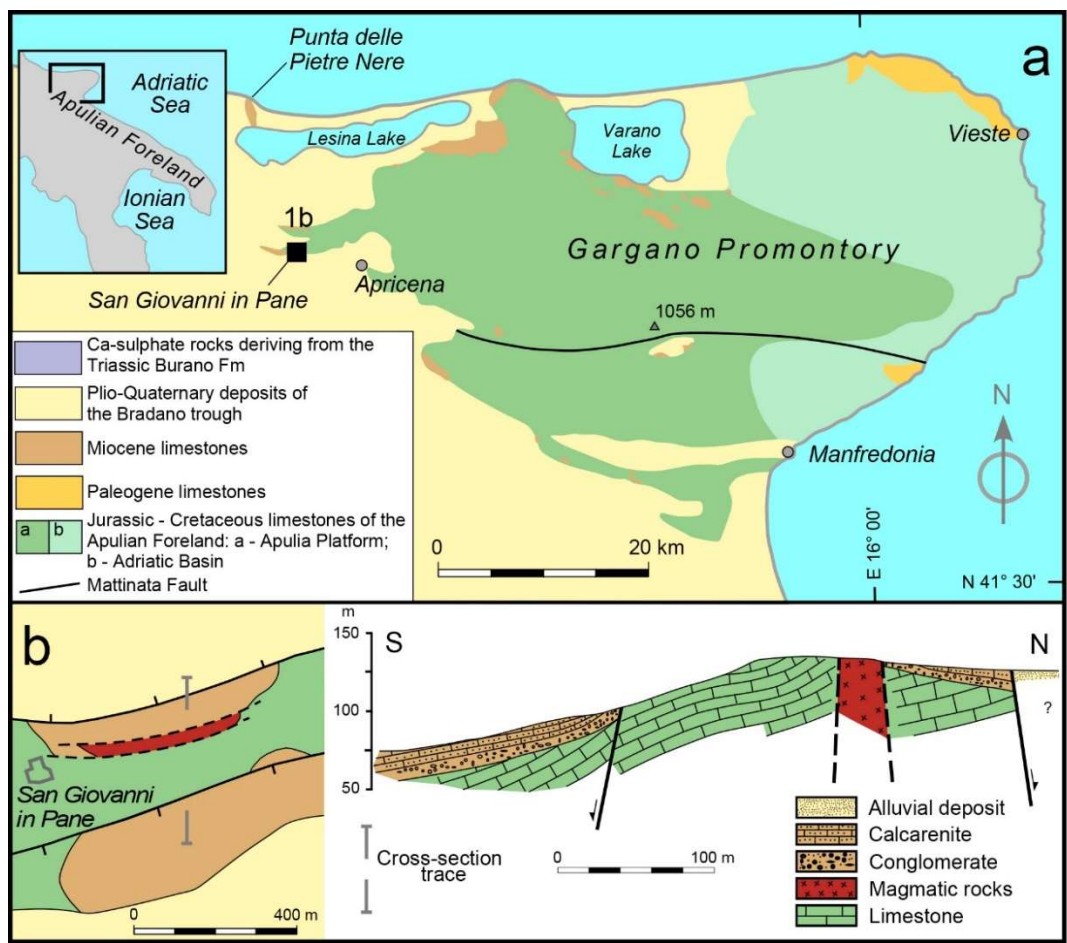

**Figure 1.** (**a**) Geological sketch map showing the location of the Punta delle Pietre Nere and San Giovanni in Pane mafic and ultramafic rocks in the context of northern Apulian foreland; (**b**) geological sketch map of the San Giovanni in Pane area (left) and related cross-section (right) modified from the study of Carella [38].

## 3. Materials and Methods

Rock composition was obtained for powders analyzed at Activation Laboratories (www.actlabs. com) in Kamloops, BC, Canada. Fusion of rock powders was performed employing a lithium metaborate/tetraborate and was followed by a digestion of the bead with weak nitric acid solution. Major oxides and trace elements concentrations were then determined by ICP optical emission spectrometry and ICP mass spectrometry, respectively. The petrographic study was carried out combining observation of thin sections at the microscope with phase analysis and identification at the SEM-EDS. A Zeiss EVO 50 XVP SEM having an EDS spectrometer with X-Max Silicon drift detector from Oxford Instruments, hosted at Dipartimento di Scienze della Terra e Geoambientali of Bari University, was used. Operating conditions were: 15 kV accelerating potential; 1-nA probe current; counting time 100 s; working distance of 8.5 mm. More refined mineral analyses were obtained by EMPA at the Dipartimento di Scienze della Terra Universita' di Milano using a JEOL JXA 8200 Superprobe equipped with five wavelength-dispersive (WDS) spectrometers. Minerals were analyzed

using 1-micron beam at 15 kV and 5-nA beam current. Natural and synthetic minerals and glasses were used as standards. In particular, omphacite USNM110607 and albite were used as a standard for Na, olivine USNM2566 for Mg, anorthite USNM137041 for Al, wollastonite for Ca and Si, K-feldspar PSU-Or1A for K, ilmenite USNM96189 for Ti, chromite USNM117075 and pure Cr for Cr, rhodonite for Mn, fayalite USNM85276 for Fe, nickeline and pure Ni for Ni and sanbornite for Ba. All standards were calibrated within 0.5% at one standard deviation. Raw data were corrected using a Phi-Rho-Z quantitative analysis program.

In addition, for one rock type, a few phlogopite single crystals were directly hand-picked from crushed rocks, in order to carry out a structural investigation. A Bruker AXS APEXII diffractometer with Kappa geometry, installed at Dipartimento di Scienze della Terra e Geoambientali, University of Bari (Bari, Italy), equipped with a CCD area detector (model 1 K SMART; 40-mm crystal-to-detector distance) and graphite-monochromatized MoKα radiation (50-kV and 30-mA operating conditions), was used for data collection.

The collection strategy was optimized by the Apex program suite by combining several ω and φ rotation scans (scan width 1.0°/frame, exposure time 10 s/frame). Data reduction, including intensity integration, correction for Lorentz and polarization effects, was done using the software SAINT-IRIX 6.28 [40]. A semi-empirical absorption correction was applied to all data using the SADABS package [41]. Preliminary lattice parameters and an orientation matrix were obtained from the xyz centroids of reflections from three sets of frames and refined during the integration process of the intensity data. Data analysis, initially carried out with XPREP software Version 6.02 [42], confirmed the space group *C*2/*m*. Subsequently, the crystal structure was anisotropically refined by a full-matrix least-squares method using the CRYSTALS software Version 12 [43] starting from the atomic parameters of Scordari et al. [44]. Scattering curves for fully ionized chemical species were used for non-tetrahedral sites, whereas ionized vs. neutral scattering curves were used for Si and O. Only reflections with $I > 3\sigma(I)$ were considered suitable for the structure refinements. Refined parameters were atomic positions, anisotropic atomic displacement parameters and the overall scale factor.

Single-crystal Raman spectra were obtained with a confocal Horiba Jobin Yvon Labram HR Evolution spectrometer at Dipartimento di Chimica, University of Bari (Bari, Italy). This spectrometer is equipped with an Olympus optical microscope, an ultra-low frequency (ULF) filter, a He–Ne laser working at 632.8 nm and a multichannel air-cooled charge-coupled device (CCD) detector. The samples were positioned under the microscope objective with the *hk0* faces, which are also cleavage planes, normal to the incident beam. The measurements were carried out at room temperature and each spectrum was accumulated three times with an integration time of 60 s to improve the signal to noise ratio. Spectral data were smoothed, and baseline corrected using the LabSpec 6 software. The Raman peak position of an unstrained silicon substrate (520.5 cm$^{-1}$) was used to calibrate spectral frequency.

## 4. Results and Discussion

### 4.1. Petrology and Classification of the Studied Rock Types

Among the blocks at San Giovanni in Pane locality, we recognized two different rock types—both characterized by an elevated density of ~3.15 g/cm$^3$. The first one has a homogeneous grain size and a mineralogical composition dominated by olivine, diopside, labradoritic plagioclase and minor alkali feldspar, with titanomagnetite and apatite as main accessory phases. The second type—the main focus of this paper—has a porphyritic texture (Figure 2a) with a more unusual feldspar-free and feldspathoid-bearing mineralogical composition, including olivine, diopside, phlogopite, nepheline, analcime with titanomagnetite and apatite as accessory phases. Olivine occurs mostly as phenocrysts with a flow-related preferred orientation (Figure 2a). Predominantly, olivine shows a skeletal texture (Figure 2b) and sometimes it is partly or completely replaced by green nontronite. Diopside was by far the dominant mineral in the rock matrix. It occurs as microlites having a length in most cases between 10 and 60 μm (Figure 2c).

Phlogopite occurs throughout the rock as brown to reddish spots. It mostly occurs in the form of oikocrysts including pale green diopside chadacrysts (Figure 2c) and opaque minerals. More rarely, a phlogopite devoid of diopside inclusions makes up a corona around olivine grains (Figure 2c). Owing to these textural features, the separation of pure single crystals of phlogopite was obtained only with difficulty.

Euhedral nepheline (Figure 2d) is not homogeneously distributed throughout the rock, but it is concentrated in thin bands (Figure 2a). Nepheline was also observed in the chilled margins of the Punta delle Pietre Nere melagabbro layered body [45]. Analcime was identified in the rock matrix by SEM/EDS analysis.

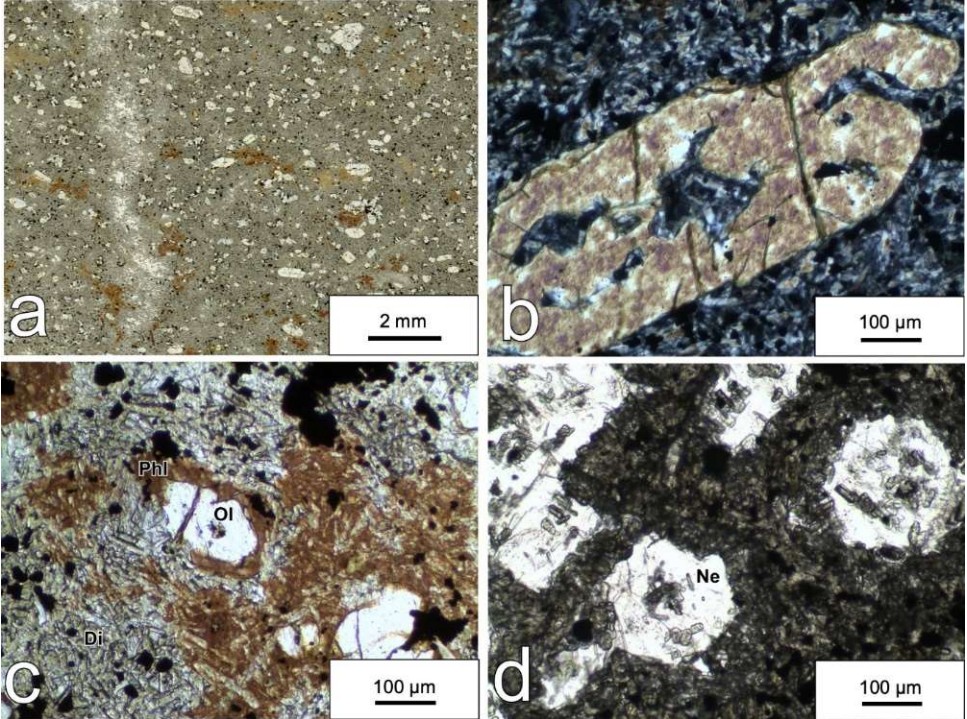

**Figure 2.** (**a**) Thin-section scan in polarized light showing the porphyritic texture of the monchiquite with olivine phenocrysts, phlogopite oikocrysts and a matrix mainly made up of diopside microlites. A vertical band enriched in nepheline can be observed on the left of the scan; (**b**) micrograph showing an olivine phenocryst with skeletal texture within the diopside-rich matrix (crossed polars); (**c**) micrograph showing oikocrysts of phlogopite enclosing numerous chadacrysts and a resorbed olivine surrounded by a thin corona of phlogopite (plane polarized light); (**d**) micrograph showing basal and longitudinal sections of nepheline microphenocrysts (plane polarized light). Phl—phlogopite; Ol—olivine; Di—diopside; Ne—nepheline.

On the basis of the mineralogical composition and rock texture, the first rock type can be classified as a melagabbro. The second type can be assigned to the lampropyhire group, with most of the characteristics typical of monchiquite. From the textural features and chiefly for the presence of skeletal olivine, microlitic diopside and acicular apatite with elongation exceeding 20:1 [46], it can be deduced that the monchiquite magma underwent a fast cooling history.

Chemical analyses for major oxides of the two rock types are provided in Table 1. Irvine and Baragar [47] classification diagram based on CIPW normative composition indicates that both rock types plot in the field of picrite basalt and ankaramite, being undersaturated in silica. The content of nepheline emerging from the CIPW norm (Table 1), indicates the highest degree of undersaturation in silica for the monchiquite, in agreement with the observed mineralogical composition.

Trace element analyses (Table 2) show that both rock types are enriched in Sr and in Ni, Cr, V and Zn. In addition—and particularly in the monchiquite—several HFSE, such as Zr, Nb, Ta, LREE are characterized by elevated contents.

REE patterns of the San Giovanni in Pane rocks (Figure 3a and Table 2) are well fractionated with an elevated CeN/YbN ratio (up to 23.8 in monchiquite) and are devoid of a significant Eu anomaly. Figure 3a shows that the patterns are similar to those of the Punta delle Pietre Nere rocks, except for the slightly lower contents of the HREE. In the multielement spider diagram normalized to the primitive mantle (Figure 3b and Table 2), overall convex-up patterns with maxima at Ta and Nb can be observed. A pronounced K negative anomaly, sharper in the monchiquite rock type, is very distinctive. Lower negative anomalies can be observed for Rb and Hf. Compared to the San Giovanni in Pane rocks, the Punta delle Pietre Nere rocks, still show the negative anomaly of K, even though with a lower intensity. Rb and, to a minor extent, Ba, Yb and Lu are higher in the Punta delle Pietre Nere samples. Instead, Sr, probably carried in large amount by apatite, seems to be more elevated in the San Giovanni in Pane samples. It can be concluded that the geochemical features of the San Giovanni in Pane rocks are similar to those of the PPN rocks, although some differences—chiefly the more pronounced K negative anomaly—appear indicative of a slightly different magma typology.

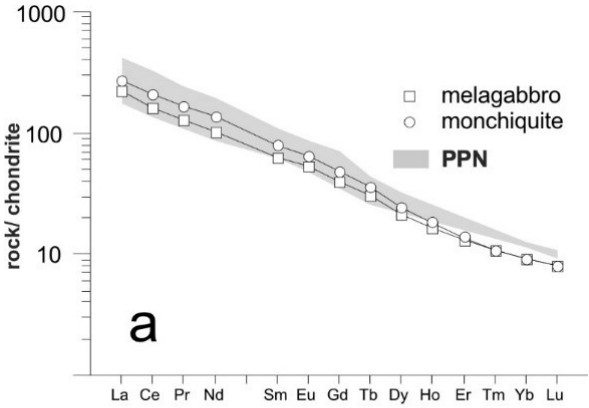

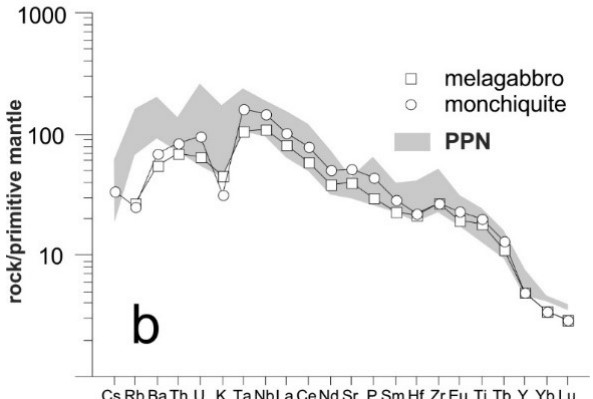

**Figure 3.** REE patterns (**a**) and multi-element spider diagram (**b**) of the San Giovanni in Pane and Punta delle Pietre Nere (PPN) rocks. Normalizing values are from McDonough and Sun [48] and Sun and McDonough [49] in (a) and (b), respectively. Patterns of PPN rocks (in gray) from analyses by De Astis et al. [35] and Avanzinelli et al. [50].

**Table 1.** Major oxide contents and CIPW normative composition of the melagabbro and monchiquite samples from San Giovanni in Pane. For norm calculation it was assumed a $Fe^{3+}/(Fe^{3+} + Fe^{2+})$ ratio of 0.30.

|  | Melagabbro | Monchiquite |
|---|---|---|
| $SiO_2$ | 43.20 | 40.00 |
| $TiO_2$ | 3.61 | 3.99 |
| $Al_2O_3$ | 9.92 | 8.99 |
| $Fe_2O_3$ | 12.80 | 13.57 |
| MnO | 0.17 | 0.18 |
| MgO | 11.60 | 13.23 |
| CaO | 12.17 | 13.52 |
| $Na_2O$ | 2.09 | 2.46 |
| $K_2O$ | 1.31 | 0.92 |
| $P_2O_5$ | 0.62 | 0.92 |
| LOI | 0.87 | 1.61 |
| tot | 98.34 | 99.39 |
| or | 8.04 | 3.34 |
| ab | 10.20 | - |
| an | 13.98 | 11.11 |
| ne | 4.56 | 11.69 |
| lc | - | 1.78 |
| di | 34.94 | 41.15 |
| ol | 13.94 | 14.84 |
| mt | 5.77 | 6.09 |
| il | 7.10 | 7.82 |
| ap | 1.48 | 2.20 |

LOI = loss on ignition; or—orthoclase; ab—albite; an—anorthite; ne—nepheline; lc—leucite; di—diospide; ol—olivine; mt —magnetite; il—ilmenite; ap—apatite.

**Table 2.** Trace element contents of the melagabbro and monchiquite samples from San Giovanni in Pane.

|  | Melagabbro | Monchiquite |
|---|---|---|
| Sc | 27 | 24 |
| V | 298 | 321 |
| Cr | 570 | 550 |
| Co | 59 | 62 |
| Ni | 260 | 330 |
| Cu | 60 | 30 |
| Zn | 110 | 150 |
| Ga | 20 | 20 |
| Ge | 2 | 2 |
| Rb | 16 | 15 |
| Sr | 794 | 1026 |
| Y | 21 | 21 |
| Zr | 278 | 283 |
| Nb | 71 | 95 |
| Ag | 1.3 | 1.3 |
| Sn | 2 | 2 |
| Ba | 358 | 456 |
| La | 52.6 | 65.2 |
| Ce | 98.4 | 130.0 |
| Pr | 12.0 | 15.7 |
| Nd | 47.4 | 63.2 |
| Sm | 9.3 | 11.7 |
| Eu | 2.94 | 3.57 |

**Table 2.** *Cont.*

|  | Melagabbro | Monchiquite |
|---|---|---|
| Gd | 7.9 | 9.5 |
| Tb | 1.1 | 1.3 |
| Dy | 5.3 | 6.0 |
| Ho | 0.9 | 1.0 |
| Er | 2.1 | 2.2 |
| Tm | 0.27 | 0.27 |
| Yb | 1.5 | 1.5 |
| Lu | 0.2 | 0.2 |
| Hf | 6.0 | 6.2 |
| Ta | 3.9 | 5.9 |
| Th | 5.5 | 6.6 |
| U | 1.3 | 1.9 |

*4.2. Chemistry of the Monchiquite Main Minerals*

Microprobe analyses were made of phlogopite and the other main phases of the monchiquite rock type (Tables 3 and 4). EMPA data for each mineral in Tables 3 and 4 represent the average over about five spots. Results indicate that olivine has a moderate compositional variation from Fo76 to Fo81 and is characterized by a significant content in CaO, up to 0.45 wt% (Table 3). The observed Ca content confirms that the olivine-melt system underwent a fast cooling, preventing the Ca loss expected during re-equilibration at progressively lower temperatures.

Diopside [51]—owing to the elevated contents in $Al_2O_3$ and $TiO_2$ (4.37 and 2.97 wt%, respectively)—is a aluminian titanian variety with an average Mg/(Mg + Fe) ratio of 0.79. In addition, the value of $Fe^{3+}$, estimated on the basis of the charge balance, tends to be relevant and in some cases higher than $Fe^{2+}$.

Nepheline composition is characterized by a maximum content of the kalsilite molecule of ~20%. In addition, it shows a low, but significant content of Ca and a silicon to aluminum ratio exceeding the value of the ideal formula (Table 3).

As concerns phlogopite, the main compositional features (Table 4) are the elevated content in $TiO_2$ and BaO (10.10 and 3.34 wt%, respectively) with an average Mg/(Mg + Fe) ratio of 0.71.

In order to investigate the compositional variability, in terms of Ti and Ba contents, a large number ($n$ = 53) of SEM/EDS analyses was also performed. Observed maximum contents in $TiO_2$ and BaO were, respectively, 10.83 and 7.39 wt%. While the variability in $TiO_2$ content is moderate (9.78 ± 0.52), the BaO content widely changes (3.73 ± 2.07). The Ba content is a maximum in the phlogopite oikocrysts and very low or zero in phlogopite coronas around olivine. The SEM/BSE image in Figure 4 shows the difference in brightness between the two phlogopite types, owing to the contrasting content in Ba.

Overall, chemical features indicate that mica can be classified as barian titanian phlogopite. In a comparison with the phlogopite found in the melasyenite of the Punta delle Pietre Nere outcrop ($TiO_2$ up to 6.33 wt% and BaO not quantified in [45]), Ti content in the presently studied mica is distinctly higher.

**Table 3.** Average composition of olivine (Ol), clinopyroxene (Cpx) and nepheline (Ne) in the monchiquite sample. Standard deviation in parenthesis.

| Oxide | Ol | Range (Min–Max) | Cpx | Range (Min–Max) | Ne | Range (Min–Max) |
|---|---|---|---|---|---|---|
| $SiO_2$ | 38.97 (0.61) | 38.32–39.95 | 48.63 (2.06) | 45.98–51.12 | 42.88 (0.36) | 42.63–43.14 |
| $TiO_2$ | 0.05 (0.04) | 0.01–0.08 | 2.97 (0.66) | 2.23–3.80 | 0.11 (0.04) | 0.08–0.13 |
| $Al_2O_3$ | 0.01 (0.01) | 0.01–0.03 | 4.37 (1.39) | 2.74–5.93 | 33.19 (1.07) | 32.36–34.02 |
| $Cr_2O_3$ |  |  | 0.01 (0.03) | 0.01–0.07 |  |  |

**Table 3.** *Cont.*

| Oxide | Ol | Range (Min–Max) | Cpx | Range (Min–Max) | Ne | Range (Min–Max) |
|---|---|---|---|---|---|---|
| FeO | 20.14 (2.28) | 17.01–22.87 | 6.27 (0.80) | 5.36–7.30 | 1.01 (0.36) | 0.62–1.40 |
| MnO | 0.42 (0.14) | 0.23–0.61 | 0.14 (0.04) | 0.10–0.20 | | |
| MgO | 41.25 (2.21) | 38.59–44.15 | 13.44 (0.91) | 12.31–14.42 | 0.23 (0.31) | 0.02–0.58 |
| NiO | 0.19 (0.07) | 0.13–0.31 | | | | |
| CaO | 0.33 (0.13) | 0.15–0.45 | 22.88 (0.66) | 21.73–23.32 | 1.28 (0.07) | 1.23–1.33 |
| Na$_2$O | | | 0.80 (0.17) | 0.67–1.10 | 15.68 (0.55) | 15.27–16.07 |
| K$_2$O | | | | | 5.39 (0.17) | 5.21–5.60 |
| tot | 101.36 (0.35) | | 99.51 (0.32) | | 99.77 (0.65) | |
| | | | | | | |
| Si | 0.99 | | 1.81 | | 2.07 | |
| $^{IV}$Al | | | 0.18 | | 1.89 | |
| $^{VI}$Al | | | 0.02 | | | |
| Ti | 0.00 | | 0.08 | | 0.00 | |
| Cr | | | 0.00 | | | |
| Fet | 0.43 | | 0.20 | | 0.04 | |
| Mn | 0.01 | | 0.00 | | | |
| Mg | 1.56 | | 0.75 | | 0.02 | |
| Ni | 0.00 | | | | | |
| Ca | 0.01 | | 0.91 | | 0.07 | |
| Na | | | 0.06 | | 1.47 | |
| K | | | | | 0.33 | |
| $\sum$O * | 4.00 | | 6.00 | | 8.00 | |
| $\sum$cat | 3.00 | | 4.01 | | 5.89 | |
| Mg/(Mg + Fe) | 0.78 | | 0.79 | | | |
| * Fe$^{3+}$ | | | 0.080 | | | |
| * Fe$^{2+}$ | | | 0.12 | | | |

* formulae calculated on the basis of $\sum_O$ = 4 and $\sum_O$ = 8 for Ol and Ne, respectively. Cpx formula calculated on the basis of $\sum_O$ = 6 and $\sum_{cat}$ = 4 with Fe$^{3+}$ and Fe$^{2+}$ estimated by charge balance method.

**Table 4.** Average chemical composition and structural formula (apfu) of the barian titanian phlogopite investigated by single-crystal X-ray diffraction (SCXRD). Standard deviation in parenthesis.

| | wt.% | Range (min–max) | | a.p.f.u. |
|---|---|---|---|---|
| SiO$_2$ | 34.88(1.68) | 33.15-36.83 | Si | 2.68 |
| TiO$_2$ | 10.10(0.67) | 9.52-10.89 | Al | 1.27 |
| Al$_2$O$_3$ | 13.96(0.45) | 13.57-14.61 | Fe$^{3+}$ | 0.05 |
| Cr$_2$O$_3$ | 0.07(0.08) | 0.01-0.19 | $^{[IV]}\Sigma$ | 4.00 |
| FeO$_{tot}$ | 11.09(1.16) | 9.85-12.52 | Ti | 0.59 |
| MnO | 0.06(0.03) | 0.04-0.10 | Mg | 1.75 |
| MgO | 15.30(0.96) | 14.20-16.25 | * Fe$^{2+}$ | 0.56 |
| BaO | 3.34(1.31) | 2.86-5.45 | * Fe$^{3+}$ | 0.09 |
| CaO | 0.09(0.03) | 0.07-0.13 | Mn | 0.00 |
| Na$_2$O | 0.67(0.02) | 0.64-0.69 | Cr | 0.01 |
| K$_2$O | 7.91(0.94) | 6.98-9.08 | $^{[VI]}\Sigma$ | 3.00 |
| F | 0.74(0.11) | 0.58-0.86 | K | 0.78 |
| † H$_2$O | 1.54 | | Na | 0.10 |
| Total | 99.75 | | Ba | 0.10 |
| | | | Ca | 0.01 |
| | | | $^{[XII]}\Sigma$ | 0.99 |
| | | | F | 0.18 |
| | | | OH | 0.79 |
| | | | O | 11.03 |
| | | | $\Sigma$ | 12 |

† from regression analysis [17]. * calculated from fit between microprobe analysis and refinement data.

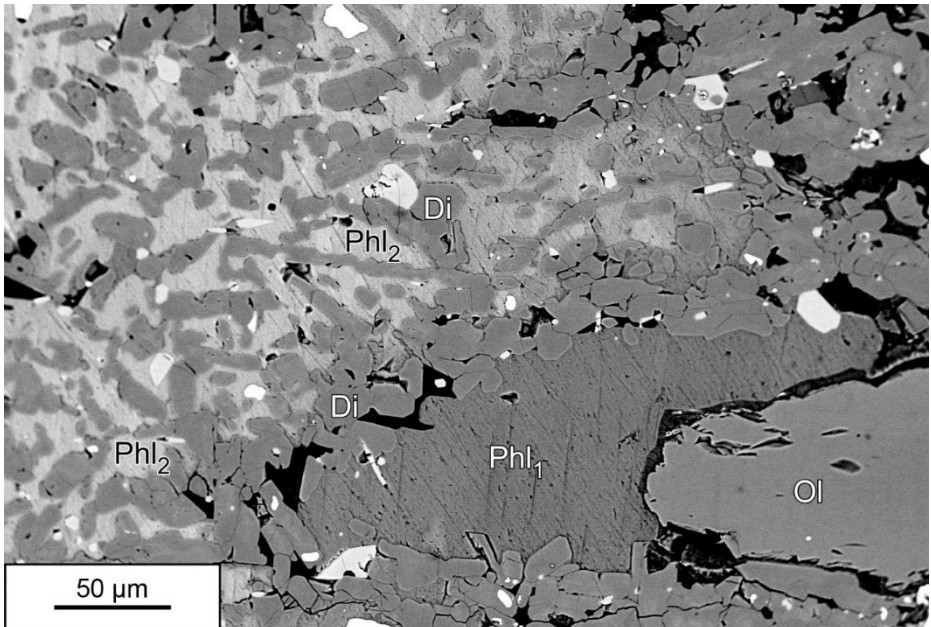

**Figure 4.** SEM/BSE image showing the difference in brightness between the phlogopite ($Phl_1$) in corona around olivine (Ol) and the oikocrysts of phlogopite ($Phl_2$) enclosing numerous chadacrysts of diopside (Di). The bright $Phl_2$ is characterized by a high content in Ba (BaO up to 7.39 wt%) whereas the dark $Phl_1$ has a low (BaO up to 1.46 wt%) to negligible Ba content. The brightest mineral phases are titanomagnetite (white) and apatite.

### 4.3. Crystallographic Features of the Barian Titanian Phlogopite

The analyzed phlogopite sample belong to the 1*M* polytype, having sharp reflections both for $k = 3n$ and $k \neq 3n$ indicating that disorder arising from stacking faults, a well-known feature in phyllosilicates, is very unlikely. The results of the structure refinement performed in space group *C*2/*m*, converged to $R_1 = 3.1$ (Table 5). Similar results were found for three other single crystals and are reported as crystallographic interchange format (CIF) files in the Supplementary Materials. The final atomic coordinates and displacement parameters are reported in Table 6, whereas selected bond distances and main distortion parameters which describe the structural features of the investigated phlogopite are provided in Table 7 and are discussed taking into account the unusual high content of Ti and Ba.

In particular, we can note marked geometric differences (bond distances and distortion parameters) between M1 and M2 octahedral site in response of high Ti incorporation. This is due to the known preference of Ti for M2 site, as it has been confirmed by several structural studies of natural biotite, e.g., [17,26,52].

Moreover, the sample is characterized by a highly distorted M2 polyhedron, as evidenced by high values of M2 bond lengths distortion, $BLD_{M2}$ (Table 7) with distance M2–O4 remarkably shorter with respect to the other two M2–O3 distances as a consequence of the strong displacement, $shift_{M2}$ (Table 7) of the M2 central cation from the geometric center of the octahedron.

The observed distortions fall in the trend (Figure 5) defined by most oxy-biotites found by other authors [17,26,27,44]. The remaining structural parameters concerning the octahedral sheet (Table 7) are consistent with those found in the literature for Ti–oxy phlogopites. The short *c* lattice parameter (Table 5) and short K-O4 distance (Table 7) confirm that the Ti incorporation is ruled by the oxy-mechanism involving the deprotonation process at O4 according to the exchange mechanism $(Mg, Fe^{2+}) + 2OH^{-1} \leftrightarrow Ti^{4+} + 2O^{2-}$. Indeed, in phlogopite where the oxy-mechanisms are important, the short *c* parameter is a consequence of the shortening of the K-O4 distance due to the decreased

coulombic repulsion between the interlayer cation and the O4 site when $OH^-/O^{2-}$ replacement occurs [53,54].

Recently, other geometric parameters were introduced in order to better discriminate the Ti-substitution mechanism involved in the mica structure. In particular, a relationship between the difference $(O3-O3)_{M1}-(O3-O3)_{M2}$ and the displacement of the O4 oxygen ($[(xO3-xO4)\cdot a]$) from the center of a octahedral hexagon defined by the O3 oxygen atoms around the O4 site was found [27,28]. These two parameters, calculated for the sample under study here, fall within the trend defined by most Ti–oxy biotites (Figure 6). We can, therefore, conclude that in our samples all the structural effects point to the Ti–oxy substitution mechanism occurring in the structure.

Other structural features characterizing our sample concern the tetrahedral and interlayer sheets. It can be noted a small tetrahedron rotation angle ($\alpha = 7.16°$). The parameter $\alpha$ describes the in-plane rotation of adjacent tetrahedra in opposite directions about $c^*$ and main adjustment mechanism to obtain congruence between tetrahedral and octahedral sheet [18]. Oxy-mechanisms, causing the opposite movement of oxygens at O3 and O4 site along the [100] crystallographic direction, determine small $\alpha$ values [54].

The $\alpha$ rotation angle is inversely related to the difference between the inner and outer distances ($\Delta$K-O) around the interlayer cation and to the interlayer cation coordination number [54]. Less rotation of the tetrahedra produces a larger size and more hexagonal shape of the tetrahedra ring which allows Ba to better fit within the ring [21] as a consequence of the preference of the divalent Ba to surround itself by as many neighbors as possible [19]. We determined the effective coordination number (ECoN) of the interlayer cation using the iterative procedure described in Nespolo et al. [55]. We found a decidedly high coordination number (ECoN = 10.72), considering that the ECoN varies from 9 to 11 in phlogopite [18].

**Table 5.** Crystal, experimental and refinement data.

| Sample | SGP-Ph2 |
|---|---|
| Crystal size (mm) | $0.630 \times 0.340 \times 0.030$ |
| Space group | *C2/m* |
| *a* (Å) | 5.3370 (1) |
| *b* (Å) | 9.2462 (2) |
| *c* (Å) | 10.1442 (2) |
| β (°) | 100.1960 (16) |
| Volume (Å$^3$) | 492.680 (17) |
| Z | 2 |
| θ range (°) | 4 to 40.2 |
| Miller index limits | $-9 \leq h \leq 9$ |
| | $-14 \leq k \leq 16$ |
| | $-18 \leq l \leq 17$ |
| Total reflections | 5859 |
| R (int) (%) | 3.08 |
| Reflections used (I > 3σ(I)) | 1134 |
| No. of refined parameters | 68 |
| [†] Goof | 1.076 |
| * $R_1$ (on F)/w$R_2$ (on F$^2$) | 0.0314/0.0355 |
| (Δ/σ) max | 0.01 |
| Δρmin/Δρmax (e/Å$^3$) | −0.39/1.09 |

* $R_1 = \Sigma||F_o| - |F_c||/\Sigma|F_o|$; $wR_2 = \{\Sigma[w(F_o{}^2 - F_c{}^2)^2]/\Sigma[w(F_o{}^2)^2]\}^{1/2}$; [†] GooF = $\{\Sigma[w(F_o{}^2 - F_c{}^2)^2]/(n-p)\}^{1/2}$.

**Table 6.** Crystallographic coordinates and equivalent displacement parameters ($\times 10^3$ Å$^2$). Square brackets indicate cations used for occupancy refinement.

| Site | x/a | y/b | z/c | Occupancy | Ueq |
|------|-----|-----|-----|-----------|-----|
| Sample | SGP-Ph2 | | | | |
| K | 0 | 0.5 | 0 | 1.0841 (8) | 30.8 |
| T[Si] | 0.07417 (6) | 0.16708 (4) | 0.22344 (4) | 0.9913 (10) | 9.4 |
| M1[Mg] | 0 | 0 | 0.5 | 0.6725 (8) | 9.2 |
| M1[Fe] | 0 | 0 | 0.5 | 0.3271 (6) | 9.2 |
| M2[Mg] | 0 | 0.34073 (5) | 0.5 | 0.6377 (8) | 12.0 |
| M2[Fe] | 0 | 0.34073 (5) | 0.5 | 0.3621 (7) | 12.0 |
| O1 | 0.3228 (2) | 0.23206 (14) | 0.16505 (11) | 1.0000 (8) | 18.6 |
| O2 | 0.0189 (3) | 0 | 0.16712 (16) | 1.0000 (8) | 18.6 |
| O3 | 0.13116 (18) | 0.16881 (10) | 0.39060 (10) | 1.0000 (8) | 10.2 |
| O4 | 0.1299 (3) | 0.5 | 0.39854 (15) | 1.0000 (8) | 12.4 |

**Table 7.** Selected bond lengths (Å) and parameters derived from structure refinement.

| Sample | SGP-Ph2 | | |
|--------|---------|---|---|
| **Tetrahedral sheet** | | | |
| T–O1 | 1.6585(11) | tetrahedral thickness (Å) | 2.246 |
| T–O1 | 1.6551(11) | BLD$_T$ | 0.325 |
| T–O2 | 1.6556(2) | volume$_T$ (Å$^3$) | 2.345 |
| T–O3 | 1.6690(11) | $\tau$ (°) | 110.33 |
| <T–O> | 1.660 | $\alpha$ (°) | 7.12 |
| | | $\Delta z$ (Å) | 0.020 |
| **Octahedral sheet** | | | |
| M1–O4 (x2) | 2.0596(14) | octahedral thickness (Å) | 2.131 |
| M1–O3 (x4) | 2.1054(9) | BLD$_{M1}$ | 0.940 |
| <M1–O> | 2.090 | ELD$_{M1}$ | 5.534 |
| | | volume$_{M1}$ (Å$^3$) | 11.93 |
| M2–O4 (x2) | 1.9901(11) | $\Psi_{M1}$ (°) | 59.35 |
| M2–O3 (x2) | 2.1269(10) | BLD$_{M2}$ | 2.461 |
| M2–O3′ (x2) | 2.0835(9) | ELD$_{M2}$ | 5.074 |
| <M2–O> | 2.067 | Volume$_{M2}$ (Å$^3$) | 11.56 |
| <M–O> | 2.074 | $\Psi_{M2}$ (°) | 58.97 |
| | | Shift$_{M2}$ (Å) | 0.068 |
| **Interlayer** | | | |
| K–O1 (x4) | 2.977(1) | Interlayer separation (Å) | 3.310 |
| K–O1′ (x4) | 3.300(1) | t$_{K-O4}$ (Å) | 3.915 |
| K–O2 (x2) | 2.980(2) | <K–O inner > | 2.978 |
| K–O2′ (x2) | 3.318(2) | <K–O outer > | 3.306 |
| <K–O> | 3.142 | $\Delta$(K–O) | 0.328 |

Notes: tetrahedral thickness: $((zO4 + 2zO3)/3 - (zO1 + 2zO2)/3)c\sin\beta$; $\alpha$ (tetrahedral rotation angle) = $\Sigma_{i=1}^{6} \alpha_i/6$ where $\alpha_i = |120° - \Phi_i|/2$ and where $\Phi_i$ is the angle between basal edges of neighboring tetrahedra articulated in the ring. $\Delta z = [z_{(Obasal)max} - z_{(Obasal)min}] c\sin\beta$. $\tau$ (tetrahedral flattening angle) = $\Sigma_{i=1}^{3} (O_{basal}\text{-}T - O_{basal})/3$.; $\Psi$ (octahedral flattening angle) = $\cos^{-1}$ [octahedral thickness/(2<M-O, OH, F>)]; BLD = bond-length distortions [56]; ELD = edge-length distortion [56]; Shift$_{M2}$ (M2 off-center shift) = $(y_{M2} - 0.33333)b$; octahedral thickness: $(1 - 2(zO4 + 2zO3)/3)c\sin\beta$; interlayer separation: $(2(zO1 + 2zO2)/3)c\sin\beta$.; t$_{K-O4}$ = (K-O4)$\cos(\beta-90)$.

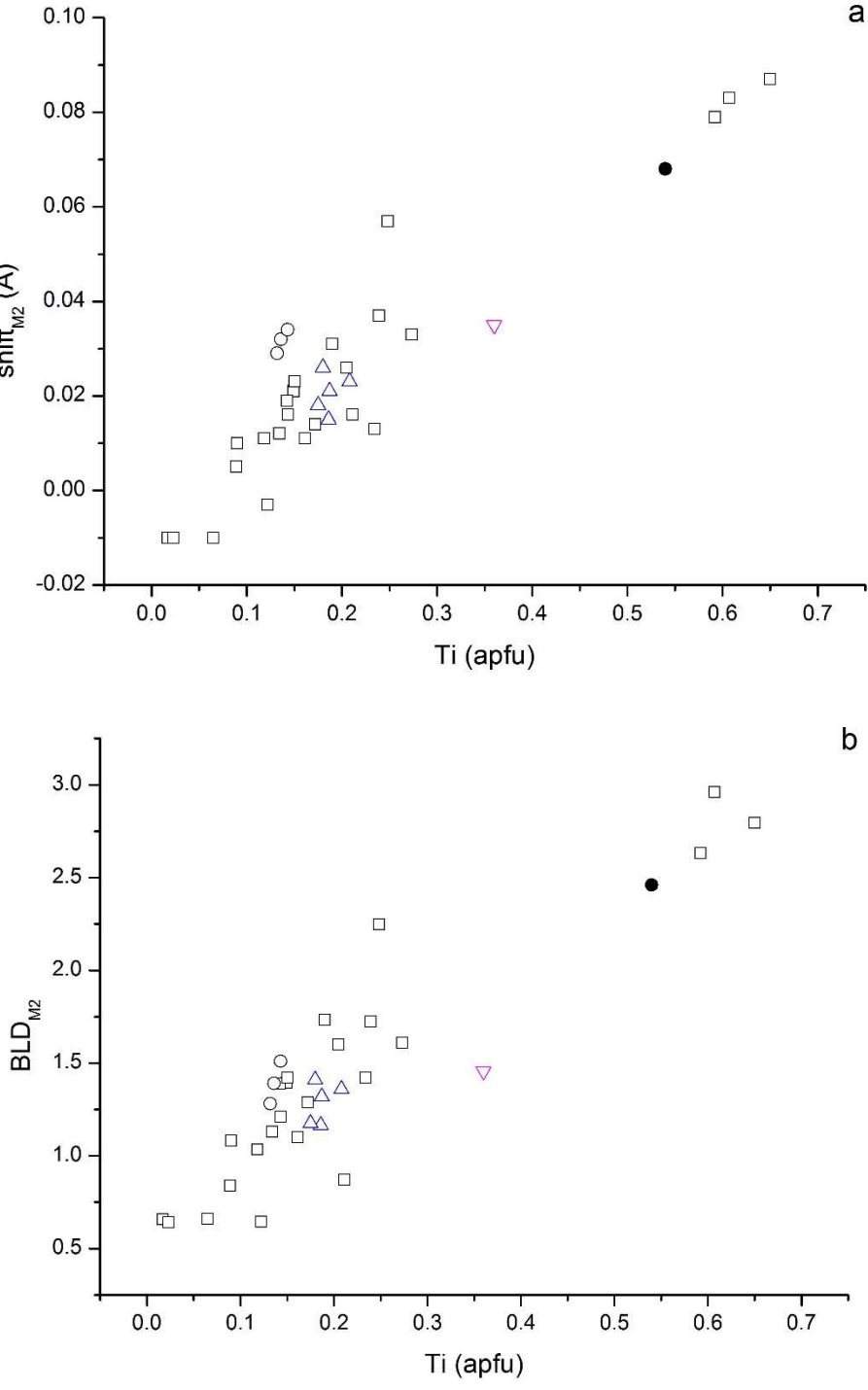

**Figure 5.** (**a**) Plot of the off-center shift of the cation within the M2 site vs. $^{VI}$Ti content; (**b**) plot of edge length distortion parameter of octahedral M2 site vs. $^{VI}$Ti content. Symbols: open squares from Cruciani and Zanazzi [17]; open circles from Cesare et al. [26]; up-pointing triangles from Scordari et al. [44]; down-pointing triangle from Schingaro et al. [27]; filled circle this study.

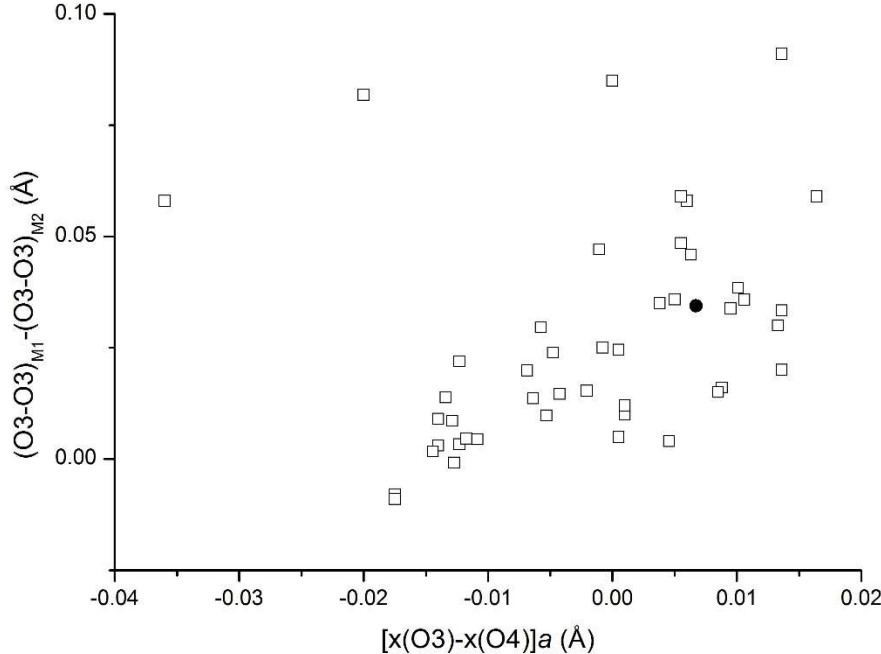

**Figure 6.** Plot of the O3 hexagon distortion vs. the O4 atom offset from the "octahedral hexagon center". Symbols: open squares—from Mesto et al. [28]; filled circle—this study.

## 4.4. Micro-Raman Spectroscopy

The Raman spectrum in the 50–3800 cm$^{-1}$ range for a Ti–Ba-rich phlogopite sample is shown in Figure 7. The spectrum shows two weak and relatively broad bands at 1090 and 890 cm$^{-1}$, a small, but well defined peak at 1006 cm$^{-1}$, a doublet of intense peaks at 773, 735 cm$^{-1}$ with a shoulder at 670 cm$^{-1}$ and wide bands at 550, 422 and 350 cm$^{-1}$. Intense bands at 190 and 150 cm$^{-1}$ plus a well-defined peak at 97 cm$^{-1}$ are in the lower part of the spectrum. Finally, a broad weak band peaked at about 3700 cm$^{-1}$ is also observed in the higher frequency part.

According to literature data for micas and other phyllosilicates, e.g., [57,58], Raman bands are generally discussed in terms of well-defined spectral regions.

The Raman peaks in the spectral region 1150–800 cm$^{-1}$ arise from the vibrational mode due to T-$O_{nb}$ stretch displacements (T = tetrahedral cation; $O_{nb}$ = non-bridging oxygen). The strongest Raman peaks in the spectral region 800–550 cm$^{-1}$ arise by the vibrational bending modes of T-$O_b$-T bonds ($O_b$ = bridging oxygen). In particular, the wide, strong peak at ~550 cm$^{-1}$ is characteristic of most Fe-bearing phyllosilicates. A correct assignment of the bands is very difficult in the spectral region <600 cm$^{-1}$, where a complex set of vibrational modes of cations in octahedral sites and in interlayer sites and the OH librational modes occur. It is generally accepted that the translational M–O (M = octahedral cation) motions give rise to the bands at about 422, 350 and 190 cm$^{-1}$. The remaining lower frequency bands at about 150 and 97 cm$^{-1}$ most likely are ascribed to I–O (I = interlayer cation) translational motions.

The Raman bands in the spectral region 3800–3000 cm$^{-1}$ are contributed by the stretching mode of hydroxyl groups (OH) coordinated with the cations (Al, $Fe^{3+}$, $Fe^{2+}$ and Mg) that occupy the octahedral sites [44,59]. The spectrum (Figure 7) shows a weak band in this region, due to oxy-mechanisms plus small amount of $F^-$ substituting the $OH^-$.

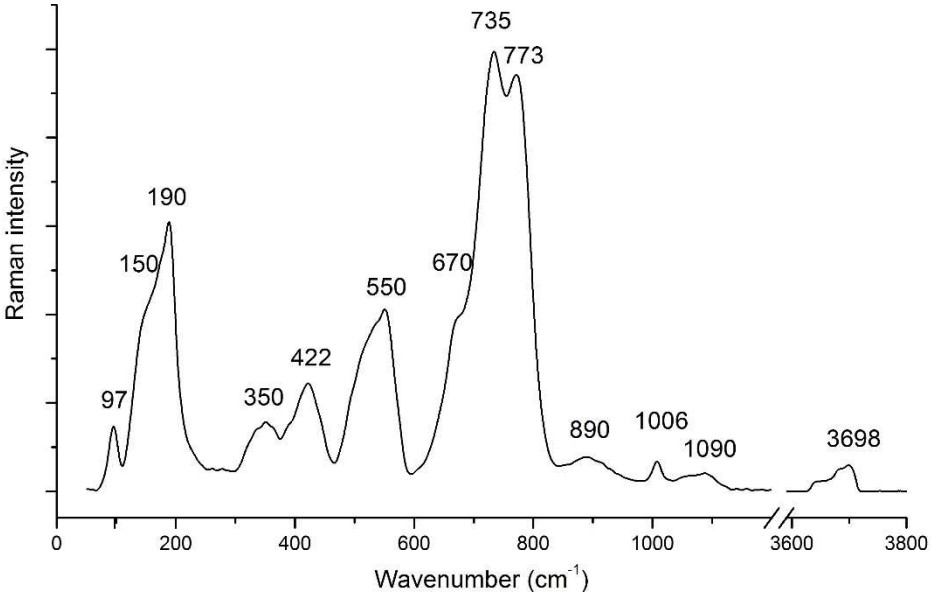

**Figure 7.** μ-Raman spectrum of the Ba–Ti phlogopite sample in the region 50–3800 cm$^{-1}$.

### 4.5. Barian Titanian Phlogopite Chemistry

The composition reported in Table 3 was obtained by combining the results from the average of the microprobe point analyses and the information obtained from single-crystal structure refinement (Tables 6 and 7). In detail, the best crystal chemical formula was reached in order to obtain (i) a good fit between the SCXRD-refined and EMPA-derived mean electron count of tetrahedral, octahedral and interlayer cation sites; (ii) an agreement between observed and calculated average bond lengths of tetrahedral and octahedral sites from ionic radii of Shannon [60].

The chemical formula was based on (O$^{2-}$, OH$^{-}$, F$^{-}$)$_{12}$ anionic charge. In this regard, an estimation of OH content was obtained from the linear regression analysis of the lattice parameter *c* vs. OH$^{-}$ [17]. This approach has demonstrated its worth, giving an OH content in agreement with H experimental determinations [26,44,54].

From the inspection of Table 3, it is worth noting that the sum of Si and Al cations is <4.0 apfu, implying that: (1) Al is confined exclusively to the tetrahedral sheet; (2) the occurrence of a small amount of another high charge cation to fill the tetrahedral site; (3) the Ti-Tschermak substitution mechanism can be ruled out, due to low Al content. As only a few single crystals were available for study, due to textural issues, it was not feasible to determine the Fe$^{2+}$ and Fe$^{3+}$ contents spectroscopically. An estimation of a Fe$^{3+}$/(Fe$^{2+}$ + Fe$^{3+}$) ratio of 0.227 was obtained by the best crystal chemical formula calculation.

The introduction of tetrahedral Ti$^{4+}$ is a controversial matter [17,28]. Although the positive correlation between Si and total Ti (Figure 8a) supports the preference of Ti for tetrahedral occupation, it is generally accepted that the presence of Ti in tetrahedral sites would decrease phlogopite stability [12]. The composition reported in Table 3 was obtained by the best match of the mean distances <M–O> and <T–O> and the mean electron counts calculated from formula molar fractions with those observed from SCXRD data. When Ti$^{4+}$ is restricted to the M sites, Fe$^{3+}$ is partitioned into both tetrahedral and octahedral sites.

Regarding the interlayer site, the investigated sample has almost stoichiometric interlayer occupancy. Tetrahedral Al is sufficient to compensate the charge imbalance due to Ba entrance. The almost perfect negative correlation obtained by plotting $^{IV}$Si + $^{XII}$K versus $^{IV}$Al + $^{XII}$Ba (Figure 8b) confirms that this coupled substitution accounts for the accommodation of whole Ba content in the structure.

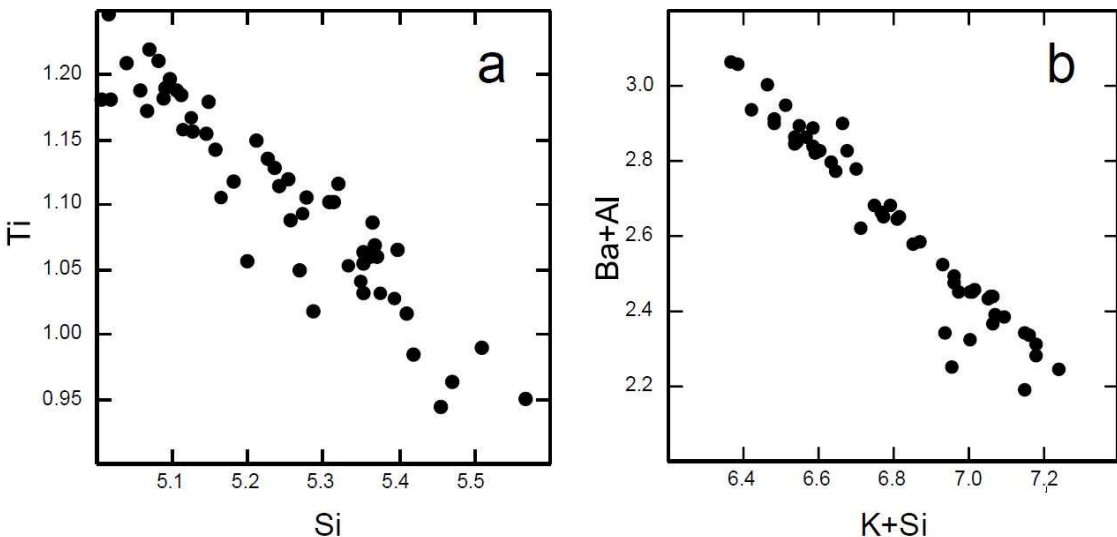

**Figure 8.** Plots showing the negative correlations between (**a**) Ti$^{4+}$ and Si and between (**b**) Ba + Al and K + Si. Data obtained from SEM/EDS analyses.

### 4.6. Conditions Favoring Crystallization of the Barian Titanian Phlogopite

The remarkable simultaneous enrichment in Ba and Ti of the studied phlogopite has been rarely observed and notably by Zhang et al. [12], Cruciani and Zanazzi [17] and Greenwood [16], in leucitites and lamprophyre. In some cases, Ba and Ti concentrations can be even higher than those observed in the presently studied phlogopite. Zhang et al. [12] and Greenwood [16] maintained that crystallization of the barian titanian phlogopite occurred from the residual melt at low pressure conditions. These conditions are easily confirmed for the Gargano Promontory phlogopite, as indicated by the textural features of the lamprophyre (Figure 3) and by the injection of the dyke in limestone (Figure 1b). Owing to the shallow level of emplacement, the lamprophyric magma was subjected to a fast cooling rate. The acicular shape of apatite and the skeletal texture of olivine are strictly related to this. In addition, the significant Ca content detected in olivine can be also interpreted as an effect of fast cooling, able to prevent re-equilibration and Ca loss at progressively lower temperatures. Quantitative estimation of T and cooling rate are possible, starting from the Ca content of olivine. Thermometers based on the partition of CaO and MgO between olivine and melt [61] and on the Ca exchange between olivine and clinopyroxene [62] allow to obtain olivine crystallization temperatures of ~1200 and 1240 °C, respectively. Then, following the approach described by Van Tongeren et al. [63], an initial cooling rate of ~110 °C/year can be estimated for the studied lamprophyric magma. This result seems consistent with a numeric thermal model. Taking into account the geological profile of Figure 1b and making some simple assumptions on the physical properties of the magma it is possible to reproduce the cooling history of the magma body. To this end, a 1D numeric thermal model in Stella® code was built. Model results (Figure 9) indicate that starting from a temperature of 1200 °C, the dyke border underwent an initial cooling rate of about 130 °C/year. Meanwhile, temperature in the dyke core decreased by less than 0.01 °C in the first year after emplacement.

Therefore, it is suggested here that a further condition favoring the genesis of barian titanian phlogopite is the fast cooling rate, especially close to the dyke margins. Here, after massive crystallization of olivine, nearly isolated pockets of residual melt with anomalous enrichment in elements such as Ba and Ti were possibly generated, thus allowing crystallization of titanian diopside microlites and of the barian titanian phlogopite. Interestingly, a role of the rapid cooling rate in the genesis of Ba–Ti-enriched phlogopite in kimberlite was recently underlined by Barnett and Laroulandie [64].

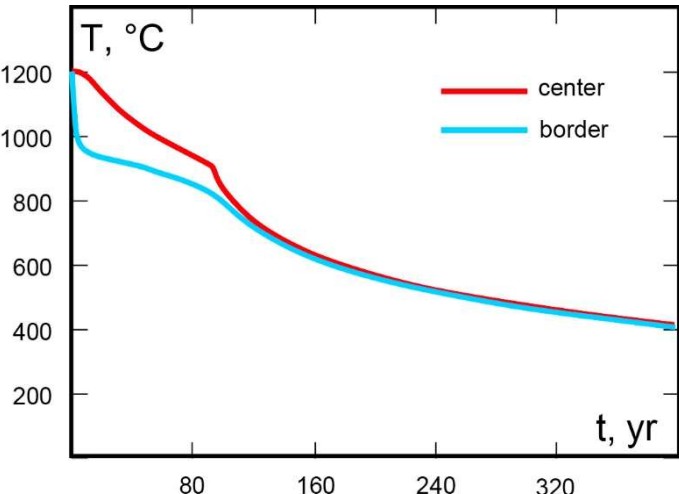

**Figure 9.** T–t diagram showing the cooling history simulated in the center and in the border of a dyke, by means of a numeric thermal model in Stella® code. The dyke has a thickness of 40 m, in accordance with Carella [38] cross-section (Figure 1b). The cooling path of the border is related to a point at a distance of 2 m from the contact with the wall rock. Initial temperatures of 1200 °C and 100 °C are assumed for magma and wall rocks, respectively. Thermal diffusivity is fixed at $8.3 \times 10^{-7}$ m$^2$/s. Finally, a value of 600 kJ/kg is assumed for the latent heat of crystallization in the T interval of 1200–900 °C.

## 5. Conclusions

The barian titanian phlogopite found in an alkaline ultramafic dyke transecting limestones of the Gargano Promontory is characterized by extremely high concentration in BaO (up to 7.4 wt% in SEM/EDS analyses) and in TiO$_2$ (up to 10.8 wt%). The phlogopite belongs to an olivine-diopside-rich lamprophyre, containing also titanomagnetite, nepheline and apatite, but free of feldspars, which has been classified as monchiquite.

Structural effects resulting from Ti incorporation in the mica structure (short *c* parameter and enhanced distortion of the M2 octahedron) are consistent with the occurrence of a Ti–oxy substitution. Raman spectrum shows a weak broad band in the principal OH-stretching region in agreement with a reduced OH content at O4 site concomitant with the Ti uptake.

The dominant Ba- and Ti-substitution mechanisms operating in the investigated sample are:

$$K + Si \leftrightarrow Ba + Al \tag{1}$$

$$(Mg, Fe^{2+}) + 2OH^{-1} \leftrightarrow Ti^{4+} + 2O^{2-}. \tag{2}$$

Phlogopite growth took place after massive crystallization of olivine in conditions of fast cooling rate (>100 °C/year). This is indicated by textural features of the monchiquite, the significant Ca content of olivine and also by a thermal modeling. Barian titanian phlogopite formed from Ti- and Ba-enriched residual melt pockets, close to the dyke margin.

**Supplementary Materials:** The following are available online at http://www.mdpi.com/2075-163X/10/9/766/s1, CIF files: SGP-Ph1; SGP-Ph2; SGP-Ph3; SGP-Ph4.

**Author Contributions:** G.V. and A.C. coordinated and wrote the paper; G.V. performed XRD and Raman experiments and data interpretation; A.C. performed SEM analyses, petrological study and thermal modeling; V.F. contributed to the "Geological setting" section. A.L. performed EMPA and ICP analyses. All authors participated in the manuscript preparation. All authors have read and agreed to the final version of the manuscript.

**Funding:** This research received no external funding.

**Acknowledgments:** We thank the two anonymous reviewers whose comments/suggestions helped improve and clarify this manuscript. This research work was supported by MIUR-Italy Dipartimenti di Eccellenza, ARTICOLO 1, COMMI 314–337 LEGGE 232/2016), by University of Bari (Italy) and by CNR-National Research Council (Italy).

**Conflicts of Interest:** The authors declare no conflict of interest.

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
