# Peer review of "Crystal Chemistry of Barian Titanian Phlogopite from a Lamprophyre of the Gargano Promontory (Apulia, Southern Italy)"

_minerals, doi:10.3390/min10090766_

Round 1
Reviewer 1 Report
Dear Authors,
I have read your manuscript on titanian phlogopite and offer comments in the attached pdf for your consideration. The science is good and I could understand all the details of the geochemistry and mineralogy. The details of single-crystal crystallography and Raman Spectroscopy are outside my knowledge, so I have been unable to assess them scientifically. However, the text reads well and appears comprehensive.
There are examples of misspelling, and instances where I had difficulty understanding exactly what was implied. Here I have offered suggestions for rewording that you might like to consider.
All the figures are useful and necessary, apart from the possibility of reducing Figure 2 to just the presently labelled options b, d, and f. Options a, c, and e either don't add any other petrographic aspects that are not covered by b, d, and f, or that have not been already illustrated in the international literature on lamprophyre petrography.
Otherwise, congratulations on an interesting paper and I look forward to seeing it in print
Regards
Reviewer 1

Author Response
Dear Reviewer,
Thank you for reading our manuscript and suggesting corrections and improvements.
In the revised version we have modified the manuscript and reworded sentences following most of your recommendations and comments.
In addition, we have reduced Figure 2, cutting off parts c and d in order to cover just the main petrographic aspects.
Concerning your observation about nepheline identification (lines 179-180), we point out that in the melagabbro nepheline was not identified, but reproduced by CIPW norm calculation in minor amount (Table 1a) with respect to the monchiquite.
Kind regards,
Reviewer 2 Report
Comments on the paper “Barian Titanian phlogopite from a lamprophyre of the Gargano Promontory (southern Italy)” by Ventruti et al. submitted to Minerals.
The paper presents a study by a multianalytical approach of barian titanian phlogopite found in an alkaline ultramafic dyke (monchiquite) intruding Mesozoic limestones of the Gargano Promontory (Apulia, Italy). Using petrographic study, electron microprobe analysis, scanning electron microscopy, single-crystal X-ray diffraction, micro-Raman spectroscopy, and ICP-MS, authors concluded about the possible origin of this rare barian titanian phlogopite.
I do not hesitate to state that the paper is based on a detailed analytical study. The results are multidisciplinary and very important for understanding of possible mechanism for formation of the barian titanian phlogopite under conditions of fast cooling rate of a magma. I have no strong objection on the data interpretation. Further are specific comments line-by-line.
Lines 2-3. I would suggest changing the title of the paper, for example, “Crystal chemistry and mechanism of formation of barian titanian phlogopite from a lamprophyre of the Gargano Promontory (Apulia, Southern Italy)”.
Line 80-81. This phrase is incorrect, since it was meant to study, and not provide a favorable environment for the phlogopite crystallization.
Lines 261. The table title does not indicate that the table presents a composition of phlogopite. I suggest ”...chemical composition and structural formulae (a.p.f.u.) of the barian titanian phlogopite...”
Line 265-266 I suggest referring to the composition of phl 2 (Table 2b). Why is composition of phl 1 not shown for comparison?
Line 306 [100] was specified incorrectly
Line 402 Does the graph show Ti (IV) or Ti total?
Line 444 In Table 2b - BaO 3 wt.%
Some typos in the text:
Line 30 remove )
Line 168 feldspatoind – remove n
Line 289 after T-oxy – whitespace
Line 345 phlogopiteite - remove ite
In conclusion, the paper is suitable for publication in Minerals after minor-to-moderate revision.
Reviewer 2
Author Response
Dear Reviewer,
Thank you for reading our manuscript and suggesting corrections and improvements.
In the revised version, we have modified the manuscript following your recommendations (below in bold).
Lines 2-3. I would suggest changing the title of the paper, for example, “Crystal chemistry and mechanism of formation of barian titanian phlogopite from a lamprophyre of the Gargano Promontory (Apulia, Southern Italy)”.
We have modified the title (“Crystal chemistry of barian titanian phlogopite from a lamprophyre of the Gargano Promontory (Apulia, Southern Italy”) privileging just the crystal chemistry that is the main focus of our study.
Line 80-81. This phrase is incorrect, since it was meant to study, and not provide a favorable environment for the phlogopite crystallization.
We agree with this suggestion and we have corrected accordingly.
Lines 261. The table title does not indicate that the table presents a composition of phlogopite. I suggest ”...chemical composition and structural formulae (a.p.f.u.) of the barian titanian phlogopite...”
corrected
Line 265-266 I suggest referring to the composition of phl 2 (Table 2b). Why is composition of phl 1 not shown for comparison?
Table 2b refers to the EMPA chemical composition of the sample investigated by SCXRD as we have better specified in its caption. Phl 1 and phl 2 (in Fig. 4) refer to phlogopites analysed just by SEM/EDS. For a comparison, in the caption of Figure 4 we have reported in addition the different BaO ranges of the aforementioned samples.
Line 306 [100] was specified incorrectly
The notation [100] indicates a specific crystallographic direction. We have made the text clearer.
Line 402 Does the graph show Ti (IV) or Ti total?
Figure 8a refers to the total Ti. We have specified it both in the figure caption and in the text.
Line 444 In Table 2b - BaO 3 wt.%
BaO amount in Table 2b refers to the average BaO content determined by microprobe analysis and is correctly reported.
Some typos in the text:
Line 30 remove )
Line 168 feldspatoind – remove n
Line 289 after T-oxy – whitespace
Line 345 phlogopiteite - remove ite
all typing errors were corrected
Kind regards,